# Mental health care in the city of Lubumbashi, Democratic Republic of the Congo: Analysis of demand, supply and operational response capacity of the health district of Tshamilemba

Erick Mukala Mayoyo[1,2,3,4]*, Bart Criel[5], Joris Michielsen[5], Didier Chuy[2,4,6], Yves Coppieters[1], Faustin Chenge[2,4]

1 Centre de Recherche en Épidémiologie, Biostatistique et Recherche Clinique, Ecole de Santé Publique, Université Libre de Bruxelles, Brussels, Belgium, 2 School of Public Health, University of Lubumbashi, Lubumbashi, Haut-Katanga, DR Congo, 3 Section de Santé Communautaire, Institut Supérieur des Techniques Médicales de Kananga, Kananga, Kasaï Central, DR Congo, 4 Centre de Connaissances en Santé au Congo, Kinshasa, DR Congo, 5 Department of Public Health, Institute of Tropical Medicine in Antwerp, Antwerp, Belgium, 6 Section de Santé Communautaire, Institut Supérieur des Techniques Médicales de Lubumbashi, Lubumbashi, Haut-Katanga, DR Congo

* erickmukala1@gmail.com

## Abstract

### Background

Integrating mental health care into the primary care system is an important policy option in the Democratic Republic of the Congo (DRC). From the perspective of the integration of mental health care in district health services, this study analyzed the existing demand and supply of mental health care in the health district of Tshamilemba, which is located in Lubumbashi, the second largest city of the DRC. We critically examined the district's operational response capacity to address mental health.

### Methods

A multimethod cross-sectional exploratory study was carried out. We conducted a documentary review (including an analysis of the routine health information system) from the health district of Tshamilemba. We further organized a household survey to which 591 residents responded and conducted 5 focus group discussions (FGDs) with 50 key stakeholders (doctors, nurses, managers, community health workers and leaders, health care users). The demand for mental health care was analyzed through the assessment of the burden of mental health problems and care-seeking behaviors. The burden of mental disorders was assessed by calculating a morbidity indicator (proportion of mental health cases) and through a qualitative analysis of the psychosocial consequences as perceived by the participants. Care-seeking behavior was analyzed by calculating health service utilization indicators and more specifically the relative frequency of mental health complaints in primary health care centers, and by analyzing FGDs participants' reports. The mental health care supply available was described by using the qualitative analysis of the declarations of the

**Data Availability Statement:** All relevant data are within the manuscript and its Supporting Information files.

**Funding:** This study was carried out with the financial support of the Institute of Tropical Medicine (ITM) in Antwerp under its fourth framework agreement 2017-2021 with the Directorate-General for Development Cooperation and Humanitarian Aid (DGD). The first author (EMM) from the School of Public Health of the University of Lubumbashi (ITM partner institution) received a Marleen Boelaert research grant (number: 901004/470) as a research fellow (12/2020 - 12/2021) for the MENTAL ESP sub-project of the CREDO program funded by the aforementioned ITM-DGD framework agreement. The funder had no role in study design, data collection and analysis, decision to publish, or preparation of the manuscript.

**Competing interests:** The authors have declared that no competing interests exist.

participants (providers and users of care) to the FGDs and by analyzing the package of care available in the primary health care centers. Finally, the district's operational response capacity was assessed by making an inventory of all available resources and by analyzing qualitative data provided by health providers and managers regarding the district' capacity to address mental health conditions.

## Results

Analysis of technical documents indicated that the burden of mental health problems is a major public problem in Lubumbashi. However, the proportion of mental health cases among the general patient population seen in the outpatient curative consultations in the Tshamilemba district remains very low, at an estimated 5.3%. The interviews not only pointed to a clear demand for mental health care but also indicated that there is currently hardly any offer of care available in the district. There are no dedicated psychiatric beds, nor is there a psychiatrist or psychologist available. Participants in the FGDs stated that in this context, the main source of care for people remains traditional medicine.

## Conclusion

Our findings show a clear demand for mental health care and a lack of formal mental health care supply in the Tshamilemba district. Moreover, this district lacks adequate operational capacity to meet the mental health needs of the population. Traditional African medicine is currently the main source of mental health care in this health district. Identifying concrete priority mental health actions to address this gap, by making evidence-based mental care available, is therefore of great relevance.

## Introduction

Mental health is a state of well-being that enables people to achieve their potential, cope with daily stress, perform productive work and contribute positively to their communities [1]. It is a neglected essential component of health in low- and middle-income countries (LMICs). However, it is estimated that at least one billion people worldwide suffer from mental health problems (depression, bipolar disorder, anxiety disorders, posttraumatic stress disorder, schizophrenia, etc.) [2, 3], which accounts for 32.4% of years lived with a disability and 15% of disability-adjusted life years (DALYs) [4].

Mental health problems constitute a huge global burden of disease, particularly in LMICs, most of which have health policies based on primary care services; however, there is a large treatment gap [5]. Studies show that 75–90% of people with mental health problems in these countries do not have access to the formal mental health care they need [6–8] for several reasons, of which the four main ones related to an insufficient annual national budget allocated to the mental health subsector, lack of integration of mental health provision into primary care in most LMICs, lack of services and stigmatization are developed below.

First, in LMICs, the share of the annual national health budget allocated to the mental health subsector remains below 1% [9]. Consequently, the availability of mental health services is very low. A study assessing the availability of services in five LMICs (India, Nepal, Ethiopia, South Africa and Uganda) showed that per every 100 000 inhabitants, there were 0.06 to 6.85 beds in inpatient mental health facilities and 0.04 to 2.70 psychiatric beds in general hospitals [10].

Second, in most LMICs, there is no offer of mental health at primary care level. However, the integration of mental health into primary care services could fill the gap in patient care. For example, in Guinea, where access to mental health care is considered extremely limited, with only five psychiatrists and 38 inpatient beds for twelve million inhabitants [11], the experience of the integration of mental health care in five health centers, all members of an NGO-driven mental health program, has proven beneficial. In primary health care (PHC) services, the proportion of patients with mental health problems seen at the general consultation increased from 2.7% to 3.9% between 2012 and 2017 [12]. In addition, this integration of mental health provision has improved the quality of care for patients treated at PHC centers, particularly in its patient-centered care dimension [13]. However, the fact that the mental health care is not integrated into PHC services in most LMICs makes the coverage of mental health services very low [8, 10].

Third, due to a lack of formal mental health provision, patients resort to alternative solutions such as traditional medicine, herbal medicine and quackery [12]. Additionally, belief in the supernatural origin of diseases, including mental illnesses, leads patients to resort to fetishing, incantation and spiritual care practices [14]. Moreover, alternative approaches such as prayer, life coaching, *yoga*, *tai chi* and mindfulness meditation are sometimes recommended in hospitals for patients with psychic or psychosomatic disorders [12, 14, 15]. In addition to the therapeutic aspect, these alternative practices offer people a fundamental basis necessary for their psychosocial well-being [14–16].

Fourth, people with severe mental illness are often stigmatized within families, communities and even health services [17]. As a result, mental health remains poorly perceived by both the general public and health personnel, exposing people with severe mental disorders and psychosocial disabilities to discrimination and other forms of serious human rights violations.

Although mental health is one of the main urban health concerns around the world [18, 19], in LMICs such as the Democratic Republic of the Congo (DRC), it is a low priority since the mental health subsector has several weaknesses, such as i) lack of funding for care and essential psychotropic drugs; ii) shortage of human resources for mental health; and iii) low availability of mental health data in the national health information system [20].

Based on 2020 World Health Organization (WHO) official estimations on the burden of mental disorders, in the DRC, the DALY (per 100 000 population) for mental and substance use disorders was 1 557.7 compared to other countries in the region such as Cameroon, Central African Republic, Congo and Gabon where the DALYs were respectively 347.7, 88.4, 85.7 and 33.7 [21, 22], and the age-standardized suicide mortality rate (per 100 000 population) was 12.41 [21]. However, the country has only 6 public psychiatric hospitals and a dozen private mental health centers with 500 beds for nearly 90 million inhabitants, almost all of which are in located cities [23]. The coverage of mental health services remains very low (less than 5%); however, mental health provision is severely lacking in primary care services nationwide [24], and only 3% of PHC services have been able to integrate mental health care supply into their activity packages [24].

At the PHC level, no protocols for the care and assessment of mental health problems or official referral and counterreferral systems between PHC and mental health providers exist [24]. A 2006 report by the WHO together with the Congolese Ministry of Health shows that at most, 1 in 10 patients with a mental health condition attending PHC are referred to a mental health professional [23]. It is estimated that only 1% of PHC providers interact with mental health professionals once a year. At most, 5% of Congolese PHC providers interact with other psychosocial care providers, such as traditional healers and spiritual care workers [23]. These 5% of providers advise patients to consult traditional medicine and spiritual medicine because mental illness is considered to be of supernatural origin [25].

Regarding the burden of mental and psychosocial disorders and the important gap in the treatment of mental health problems around the DRC, the national mental health policy recommends integrating mental health care packages into the existing health services at the district level and assuring continuity of care [26]. Aligning with this national recommendation, the authorities of the health district (health zone, in the DRC) of Tshamilemba along with their development partners considered integrating mental health into the health services of this district based in the city of Lubumbashi in the Haut-Katanga province. It should be noted that for a total of 283 000 inhabitants, this health district does not currently have a dedicated psychiatric bed, nor does it have psychiatrists, psychologists, psychiatric nurses, or social workers trained in mental health.

To date, in the health district of Tshamilemba, no attempt has been made to integrate mental health into health services. Based on a study in the semirural district of Kinshasa by Mambanzi et al. [27], we could expect the gap in conventional mental health services to be filled by traditional medicine and houses of worship, despite the mistrust of intellectuals and biomedical practitioners and their lack of effective integration into the health care system.

To understand the mental health care situation in the health district of Tshamilemba, this study i) analyzed the existing demand for mental health care expressed by the populations throughout the assessment of the burden of mental health problems and care-seeking behavior, ii) described the available mental health care supply at the time of the study, and iii) critically examined the district's operational response capacity to address mental health. The findings of this study can inform the policy design for integrating mental health into primary care in other urban health districts in the DRC and other LMICs.

## Methods

### Study context

The Congolese health system is organized into 3 levels: the central level, the provincial level and the operational level. The central level is responsible for defining policies, strategies, standards and guidelines. It supports provinces by providing advice, monitoring compliance and following up the implementation of the health development plan. The provincial level ensures the application of health policy and the translation of such policy and the related strategies and guidelines into operational instructions and technical sheets to facilitate their implementation in the form of actions in the health districts [28]. The operational level is responsible for implementing health policy, which is based on the PHC strategy [29].

This study was carried out at the operational level in the health district of Tshamilemba in urban areas of Lubumbashi. The 2020 consolidated operational action plan of the health district of Tshamilemba indicates that this district, which covers an area of about 42 km$^2$ and a population of 283 000 people, is divided into 13 health areas; counts 59 private clinics including for-profit (or commercial) and non-profit (e.g. confessional) facilities, and only one public primary care facility i.e., Tshamilemba PHC center, covering about 12 000 people. This clearly illustrates the huge (and problematic) extent of privatization of the first level of health care in the city of Lubumbashi in general, and in the health district of Tshamilemba in particular.

The Tshamilemba PHC center was established in 2012 and hosts training and research activities along with health services. It tests innovative experiments in the organization of PHC in the DRC. Because of its status as a learning and research site and the fact that it is the only public PHC service in the health district of Tshamilemba, this center was chosen as the site at which to implement a project for the integration of mental health care. In 2021, the analysis of activity reports of the Tshamilemba PHC center shows that 143 cases of tuberculosis, 433 cases of HIV/AIDS, 23 cases of diabetes and 34 cases of arterial hypertension were detected and

managed. It is recognized that these chronic diseases can evolve into comorbidities with mental health problems. The analysis of activity reports also reveals an increase in the rate of utilization of general curative consultation from 0.26 new cases (NCs) per year in 2019 to 0.38 NCs per year in 2021.

## Study design

A multimethod cross-sectional exploratory study was conducted to analyze the baseline mental health situation in urban areas of Lubumbashi, particularly in the health district of Tshamilemba. This cross-sectional descriptive and content analysis study provides useful information for the development of a strategy for integrating mental health in urban areas of the DRC.

## Data sources and collection

Data were collected from three sources: (i) technical documents, (ii) key stakeholders on the mental health situation and (iii) residents of the health district of Tshamilemba. The technical documents (i.e., routine health information system canvas, consolidated district operational action plan, consultation registers, PHC centers' activities reports) are from the health district of Tshamilemba. These documents were provided by the authorities upon request of the principal investigator (EMM). In addition, a soft copy of the document of health sector budget forecasts was shared by a senior official of the Ministry of Health. In total, 19 documents were selected that were directly related to mental health in the DRC and written in either French or English. These documents allowed us to gather quantitative and qualitative data (Table 1). The documentary review was conducted by EMM and assisted by DCK (a local researcher from the health district of Tshamilemba) for two months, from December 8, 2020, to February 10, 2021. EMM together with DCK decided on the documents to be included according to whether they considered the documents relevant to the study depending on completeness, year of writing, and language of publication.

Of all the documents reviewed, only the consultation registers of the Tshamilemba PHC center contained data likely to reveal the identity of the patients seen in general curative consultation. To ensure that this patient privacy data was kept anonymous and confidential, it was collected by a single local researcher (DCK), who is also a health care provider and the managing director of the same center. In the consultation registers, DCK extracted only the data useful for research (i.e., age, sex, complaints, diagnosis), which were transcribed into an Excel file prepared for this purpose. None of the data likely to reveal the identity of the patients (i.e., name, physical address) were extracted from the registers.

**Table 1. Documents consulted and type of data collected.**

| Documents | Number | Year of production | Sources | Data extracted on: | Data type |
|---|---|---|---|---|---|
| Health Sector Budget Forecasts | 1 | 2022 | Ministry of Health | • Financial data of the mental health subsector | Quantitative |
| Consolidated District Operational Action Plan | 1 | 2020 | District management team of Tshamilemba | • Health demographics<br>• Available resources | Qualitative and quantitative |
| Consultation registers | 3 | 2020 | District health services information system | • Patient's health complaints at the medical consultation<br>• Diagnostic presumptions | Qualitative and quantitative |
| Health centers activities reports | 2 | 2020, 2021 | | • Use of services<br>• Resources available | Quantitative |
| Routine Health Information System canvas | 12 | 2021 | | • Epidemiological data on mental disorders<br>• Availability of psychotropic drugs | Quantitative |

The quantitative data extracted from the reviewed documents were saved in an Excel file. The qualitative data were recorded in a notebook prepared for this purpose, copied into a soft file and saved on a laptop whose password was kept by the principal investigator.

Key stakeholders (including doctors, nurses, managing directors, district medical officers, community health workers, community leaders and health care users) in the mental health situation were invited to participate in focus group discussions (FGDs) [30]. These key stakeholders, constituting the convenience sample, had to meet the following criteria: i) live and/or work in the health district of Tshamilemba; ii) be at least 18 years old on the day of the survey; iii) agree to participate in the study; iv) be able to express themselves in French and/or Kiswahili; and v) declare that they have familiarity with the domain of mental health, even if only empirical.

Table 2 presents a summary of the characteristics of the participants in the FGDs. We organized five FGDs in the first quarter of 2021 in the health district of Tshamilemba. The groups were formed by taking into account cultural (living environments, proximity between actors, traditions), administrative (function and role of health professionals) and economic (allotted time, funds available, participant availability) imperatives [30]. Depending on who accepted the invitation to participate in the study, heterogeneous groups of 9 to 11 participants were formed according to their professional profiles. We formed heterogeneous groups because certain socioprofessional profiles (e.g., health care users) were underrepresented because they were reluctant to participate in face-to-face FGDs due to the emergence of COVID-19. To guarantee that everybody was able to raise his or her voice and to encourage the silent ones to speak, the moderator used two techniques: the round table and distribution of speech methods [30]. Moreover, to temper dominant speakers, the moderator kindly avoided allowing them to speak for a second time for the same question.

To meet potential participants, an invitation accompanied by an information letter was sent to 90 persons seven days before the day scheduled for the meeting through the channel of

**Table 2. Summary of characteristics of participants in the FGDs (N = 50).**

| #FGDs | Meeting places | Number of participants | Gender | | Socioprofessional profile of participants (Number) | Years of experience (Min.–Max.)[a] |
|---|---|---|---|---|---|---|
| | | | M | F | | |
| FGD1 | Tshamilemba PHC center | 11 | 7 | 4 | Doctors (3)<br>Nurses (6)<br>Managing director (1)<br>Pharmacy Assistant (1) | 3–12 |
| FGD2 | Kabetha PHC center | 10 | 4 | 6 | Doctors (4)<br>Nurses (6) | 1–5 |
| FGD3 | District Central Office | 9 | 6 | 3 | District Medical Officer (1)<br>Managing director (1)<br>Nurses (5)<br>Doctors (2) | 4–11 |
| FGD4 | Community | 11 | 8 | 3 | Nurses (2)<br>Community health workers (5)<br>Community Leader (1)<br>Health care users (3) | 2–6 |
| FGD5 | Community | 9 | 4 | 5 | Nurses (2)<br>Community health workers (2)<br>Community Leaders (2)<br>Health care users (3) | 1–7 |

FGD, focus group discussion; PHC, primary health care; M, male; F, female; Min, minimum; Max, maximum.

[a]This data concerns only health professionals

community health workers based in the respective health areas. This letter included the objectives and representatives of the study, the FGD method, the confidentiality of the anonymity of the results, the overall description of the theme to be addressed and the practical arrangements for the meetings (place, date, etc.). Two days before the meetings, we tried to obtain verbal or telephone confirmation of those who would be present; 25 persons declined the invitation. The wish to avoid physical contact with other people during the COVID-19 pandemic and level of personal convenience were the two main reasons for declining. We developed the interview guide focused on three main themes: 1) demand for mental health, including the burden of mental health problems and care-seeking behavior, 2) mental health care supply available at the time of the study, and 3) the district's operational response capacity to address mental health. This last theme was discussed mainly with health professionals and community health workers. The opinions of health care users were solicited to confirm (or not) the statements of professionals. The interview guide (see S1 File) was developed in French. Once the questions were translated into the national language (Kiswahili), they were pretested in a neighboring health district to ensure that these questions could be understood by all participants.

After training the male and female investigators (MLM, MR, MTR) on the study protocol, interview tools, note taking, and observation and under the supervision of the principal investigator, the face-to-face discussions were organized and moderated by an independent researcher (public health physician, healthcare provider in a private clinic) (AKM) chosen for his expertise in conducting surveys. The FGDs were held at the PHC centers, at the central office of the health district and in the community. The interviews—which lasted an average of 1 h 45 minutes (±15 minutes)—were conducted during the day, between 8 a.m. and 5 p.m. Prior to study commencement, the interviewers introduced themselves to the participants in order to establish a relationship of trust. The discussions—in which only participants and researchers took part—were audio-recorded and the field notes made during and after FGDs by two research assistants (TKK, VIM) both doctors trained in health science and survey techniques, who originated from the city of Lubumbashi and have knowledge of the study context. During data collection, the main difficulty encountered was the restriction on gatherings of people imposed by the provincial government to limit the spread of COVID-19. Data saturation justified limiting the sample size to 50 participants.

The residents of the health district of Tshamilemba were the third source of data. A random sample in clusters was formed to carry out a household survey. The sample size was calculated for the two clusters at 769 subjects by applying the following formula [31]: n = $Za^2$ * c * (pq)/($d^2$), where: $Za^2$ = $1.96^2$; c = 2 (number 2, means clusters factor used to minimize the effect of assumed heterogeneity of the study area); p = 0.5 and q = 0.5 (as the prevalence of mental health problems is not known in the study setting) and $d^2$ = $0.05^2$. A total of 591 participants responded, for a response rate of 76.9%. Participants had to meet the following criteria: i) live in the health district of Tshamilemba; ii) be at least 18 years old on the day of the survey; iii) agree to participate in the study; and iv) be able to express themselves in French and/or Kiswahili.

The registration of participants was performed by a door-to-door strategy conducted by community health workers.

The survey questionnaire (see S2 File) we used, was previously pretested with a sample of 15 participants selected from a neighboring health district. This questionnaire included questions that aimed to explore the perceived and experienced psychosocial and mental health problems. Participants also gave their opinions on the possibilities offered by the local health system to respond to their requests for care.

## Definitions of main mental health disorders categories used

In this study, we used the definitions presented by the National Mental Health Program of the DRC in its mental health module of the NHIS (National Health Information System). In this module, the Program refers to the WHO mental health Gap Action Program (mhGAP). The symptomatic approach has been adopted to clearly define these concepts, in order to facilitate their understanding by all non-specialists (e.g. community health workers, family members, primary care providers) participating in the study. These definitions are presented in Box 1.

---

### Box 1. Definitions of main mental health disorders

**Anxiety disorders:** in the past 6 months, regular (i.e. more often than not) experiencing disorders in which anxiety and concerns are associated with at least three of the following six symptoms: restlessness or feelings of excitation or nervousness, fatigue, difficulties in concentration, irritability, muscular tension, sleeping problems and restlessness. **Stress disorders:** patients experience fear when facing stressful events, feeling of helplessness, horror, fright, reliving traumatic memories (scenes, images); avoidance of traumatic memories and hypervigilance following adverse or traumatic situations. **Depression:** A disorder in which the patient experiences persistent sadness or depressed mood, fatigue, sleep problems, decreased energy and suicidal thoughts, anxiety, loss of interest or pleasure in normally enjoyable activities for at least 2 weeks. **Substance use and related disorders:** Disorders occurring in a person apparently under the influence of a psychoactive substance, manifested by, for example, lack of energy, agitation, inability to sit still, inarticulate speech; or signs indicative of substance use such as injection marks, skin infection, disordered appearance; requesting a prescription for sedative medication (sleeping pills, opioids); having financial difficulties or criminal problems; or difficulty with work, at household level, or in habitual social activities. **Suicidal thoughts and attempts:** Disorders in which patients experience extreme hopelessness and despair; with (past or present) thoughts, plans, or acts of self-mutilation, suicide attempts, and any other condition associated with chronic pain or extreme emotional distress. **Neuroses:** Disorders in which patients experience multiple physical complaints, without apparent medical explanation or physical cause, and that do not correspond to a known illness; experience of fear without apparent reason, nervousness, difficulties in living in community. **Psychoses:** Disorders in which patients exhibit incoherent or irrelevant speech; delusions (unreal ideas believed to be true); hallucinations (hearing voices and seeing things that do not exist); feelings of withdrawal, restlessness, disorganized and aggressive behavior, fixation on false beliefs not shared by others in the person's culture, feeling that one's thoughts are being stolen, or imposed; tendency to isolate oneself and neglect habitual tasks and responsibilities related to work, school, domestic or social activities, with lack of awareness that one has mental health problems.

---

## Data management and analysis

After collection, the quantitative data compiled in the predefined data recording grid were entered into an Excel file and then processed by the principal researcher (EMM). Each of the two researchers chosen (EMM and DCK) independently assessed the scientific value of the information collected, which enabled them to obtain data deemed relevant. For these data,

frequency measures (rate, absolute frequency, proportion, etc.) were computed. Clearly, in order to understand the extent of the burden of mental health problems and health care-seeking behaviors, we calculated the morbidity indicator (proportion of mental health cases), the relative frequency of mental health complaints reported by patients in PHC centers and health care utilization indicators.

To avoid the loss of the information collected and secure confidentiality, the qualitative data collected were stored on two computers chosen by the principal researcher. A unique digital folder was created and copied to each of the two computers and then password protected. The principal researcher was responsible for managing these data and analyzing them. The computer media (laptop and flash disk) containing the transcribed data, including the NVIVO file and the quantitative data, were kept in a locked cabinet to which only researchers (EMM and DCK) had access.

During transcription, the respondents' identifiers were removed, and data were encrypted and managed anonymously by the researchers. The existing link between the code and the participants was removed, and the encryption key was saved in a separate file.

All verbatim transcripts were entered and stored in an NVIVO database for analysis. The quality and accuracy of the transcriptions were checked by the principal investigator (EMM) by randomly listening to a few recordings, and the transcriptions were then corrected if necessary.

Following the first transcriptions, clarifications were sought from the independent researcher (AKM) who had moderated the discussions, which allowed us to ensure that the questions were correctly asked and answered. The research assistants (TKK and VIM) separately read the transcripts and developed codes according to the three central themes identified in advance: 1) burden of mental health problems, 2) mental health care supply, and 3) district's operational response capacities. The coding process was supervised by two members of the research team (EMM and DCK). A thematic analysis of the data was performed using NVIVO, and the key themes and subthemes from the interviews were used to structure the study findings. After data analysis, participants were invited to provide feedback on the findings.

For this research, to understand the complexity of the baseline mental health situation in the city of Lubumbashi, two triangulation methods were used, namely, the triangulation of sources, which consisted of analyzing the different documents (technical documents and documents of health sector budget forecasts) and key stakeholders (doctors, nurses, managing directors, community health workers, etc.), and the triangulation of data collection methods, which consisted of combining the two collection methods (documentary review and FGDs). The combination of triangulation methods applies when using the multiple methods approach to understanding complex situations [32]. The use of the aforementioned triangulation methods enabled us to reach, at the end of both the documentary review and the interviews held with the key stakeholders on the mental health situation in Lubumbashi, the 'empirical' saturation of the data.

## Ethical considerations

The protocol for this study was approved by the medical ethics committee of the University of Lubumbashi (No. UNILU/CEM/034/2021) and the Institutional Review Board of the Institute of Tropical Medicine-Antwerp (IRB/AB/AC/022/1468/21). The study design was discussed with the district management team, which gave its permission, after a favorable opinion of the provincial health division of Haut-Katanga, to implement the project.

Both the interview guide and questionnaire we used were anonymous. The data extracted from the documents were already aggregated and anonymized. In the consultation registers,

data likely to reveal the identity of the patients, such as names and physical address, were not collected. This made the collected data strictly anonymous for further data management steps (i.e., processing and analyses). Participation in the interviews was voluntary, and all targeted individuals freely consented to their participation.

Written informed consent was sought and obtained from each participant after reading the contents of the information letter. Before any recording and/or note-taking, the research assistants again requested permission from the participants.

Members of the research team (interview moderator, research assistants, coders and data analyst) who participated in the collection, coding and/or data analysis were trained on ethical aspects of research (respect for privacy, confidentiality, etc.), which they have undertaken to respect. Participants were informed that recordings and transcripts would be kept for a maximum of two years and then destroyed after the study results were published.

## Results

### Characteristics of household survey participants

Overall, 591 people participated in the household survey. The average age of the participants was 36.6 (SD ±11.7). Male participants were overrepresented (55.2%). Household heads represented 72.4% of the sample. A total of 249 (42.1%) participants did not have a paid job, and 294 (37.9%) worked in the liberal sector.

### Demand for mental health care: Burden of mental health problems and care-seeking behavior

Respondents in the FGDs stated, in various ways, that mental health problems exist and represent a heavy social burden for the community and are a public health concern in the health district of Tshamilemba. Some participants stated during the FGDs that

*'We have family members with this type of disorder [namely, mental illnesses]'* [FGD4].

They expressed the importance of mental health problems in the following terms:

*'The problem of mental disorders is there; it is emerging. It has become very serious, especially with the outbreak of the COVID-19 pandemic. The [seriously] mentally ill threaten the tranquility, and even the life of those around us; they are very numerous in our communities, and many of them are roaming around the city of Lubumbashi. Mental health conditions affect various categories; all our health areas are concerned. . .'* [FGD1].

In addition, health care providers recognized that they receive and treat people with mental health problems: *'We have treated those patients with mental health problems in our health services'* [FGD1].

The care utilization indicators, resulting from the review of the selected technical documents, are presented in Table 3 below. From these results, we note that in 2020, of all general curative consultations carried out at the Tshamilemba primary care center, the proportion of patients with mental health problems out of all curative consultations was 5.3%. As well, the rate of utilization of curative mental health care at the general curative consultation was 14 new cases (NCs) per 1 000 inhabitants per year.

Although it is a neurological condition, epilepsy is frequently encountered in PHC centers. Out of a total of 3 041 patients seen at the Tshamilemba PHC center in 2020, there were 40 cases of epilepsy, representing a proportion of 1.3%.

**Table 3. Indicators of utilization of curative care at the Tshamilemba PHC center in the health district of Tshamilemba in 2020.**

| Indicators of utilization of the curative consultation in 2020 | Statistics |
|---|---|
| Population of the health subdistrict of Tshamilemba (estimated in 2020) | 11 975 |
| Number of NCs at the curative consultation at the Tshamilemba PHC center in 2020[a] | 3 041 |
| Rate of utilization of the general curative consultation at the Tshamilemba PHC center (NCs/inhab/year)[1] | 0.25 |
| Number of NCs with mental health problems seen at the curative general consultation at the Tshamilemba PHC center | 162 |
| Proportion of cases consulting for a mental health problem[b] out of all recourse of general curative consultation[2] | 5.3% |
| Rate of utilization of curative mental health care estimated at the general curative consultation (NCs/inhab/year)[3] | 0.014 |

[a] Based on our count

[b] Excluding epilepsy

[1] Numerator: Number of new cases (NCs) at the curative consultation at the Tshamilemba PHC center in 2020; Denominator: Population of the health subdistrict of Tshamilemba (estimated in 2020) (n = 11 975).

[2] Numerator: Number of NCs with mental health problems seen at the curative general consultation at the Tshamilemba PHC center; Denominator: Total number of NCs at the curative consultation at the Tshamilemba PHC center in 2020 (n = 3 041).

[3] Numerator: Number of NCs with mental health problems seen at the curative general consultation at the Tshamilemba PHC center; Denominator: Population of the health subdistrict of Tshamilemba (estimated in 2020) (n = 11 975).

During the discussions, providers of one health service stated the following:

'From a monthly average of 250 new cases seen in curative consultations, we receive up to 30 patients with mental health problems and/or complaints such as insomnia and drug addiction.' [FGD1].

The providers interviewed said that they noticed a gradual increase in mental health complaints at the general curative consultation during 2020. The documentary review confirmed this increase (Fig 1). For example, in December 2020, out of a total of 312 patients admitted to the curative consultation at the Tshamilemba PHC center, 127 reported only mental complaints, 131 patients presented only physical complaints and 54 patients reported physical complaints associated with mental complaints.

The psychosocial consequences of mental health disorders are perceived in the living environments of participants. During the interviews, they mentioned some frequently observed consequences in the following terms:

'Mentally ill people engage in self-aggressive acts, even suicide, and in heteroaggressive acts, even homicide. They disrupt public order, offend against public morals, and engage in begging. They are seen wandering around the city, even adopting a life on the streets (homelessness). They destroy family or community property. They stigmatize and shame their families, who end up abandoning them and even wishing them death. . .' [FGD2].

During the household survey, we explored the extent of the main mental health disorders felt/observed by respondents (n = 591). The corresponding results are shown in Table 4.

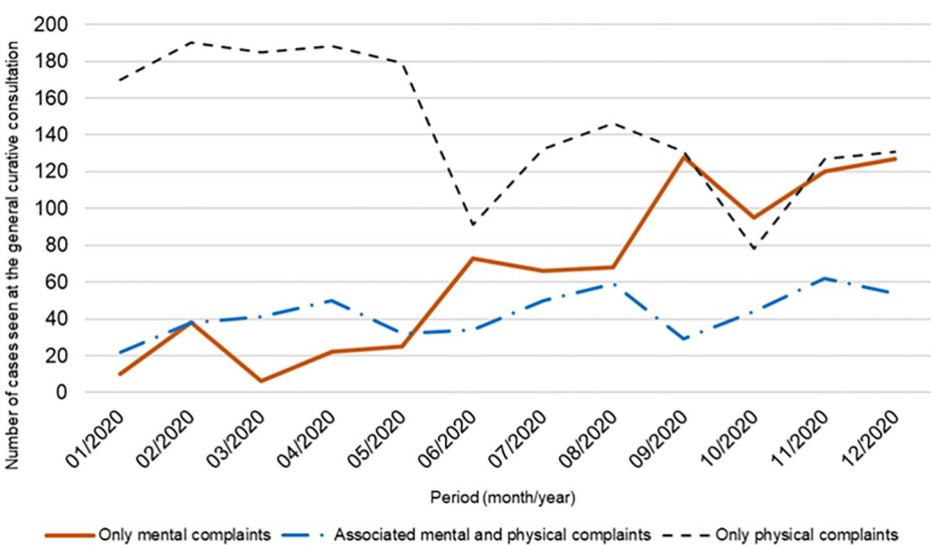

**Fig 1. Evolution of mental complaints at the general curative consultation at the Tshamilemba PHC center in 2020 (data source: Consultation registers).**

Of the 591 participants who responded to the questionnaire, 294 (49.7%) reported mental health problems either in themselves or in a family member. Of those who reported a mental health disorder (n = 294), the majority of participants and their family members were affected by stress disorders (including posttraumatic stress disorders) (34.0%; 95%CI: 30.2–37.8); substance use and related disorders (33.7%; 95%CI: 29.9–37.5); depression (27.6%; 95%CI: 24.0–31.2); and neuroses (18.7%; 95%CI: 15.7–21.9).

## Mental health care supply available

Similar to many other urban health districts across the country, mental health has not yet been integrated into PHC in Lubumbashi. The mental health provision that exists in the city of

**Table 4. Distribution of mental health cases reported by respondents in the household survey (n = 591).**

| Mental health disorders reported by respondents | Statistics | | |
|---|---|---|---|
| | n | Percentage | 95%IC |
| Did not report mental health problems[1] | 297 | 50.3 | [47.6–52.8] |
| Did report mental health problems[2] | 294 | 49.7 | [47.1–52.3] |
| Anxiety disorders[3] | 12 | 4.0 | [2.4–5.6] |
| Stress disorders[3] | 100 | 34.0 | [30.2–37.8] |
| Depression[3] | 81 | 27.6 | [24.0–31.2] |
| Substance use and related disorders[3] | 99 | 33.7 | [29.9–37.5] |
| Suicide thoughts and attempts[3] | 19 | 6.5 | [4.7–8.3] |
| Neuroses[3] | 55 | 18.7 | [15.7–21.9] |
| Psychoses[3] | 7 | 2.4 | [1.2–3.6] |

[1] Numerator: Number of respondents who did not reported a mental health problem (n = 297); Denominator: Total respondents (N = 591).

[2] Numerator: Number of respondents who reported mental health problems according to the symptoms felt/observed (n = 294); Denominator: Total respondents (N = 591).

[3] Numerator: Cases reported according to the symptoms felt/observed; Denominator: Total respondents (n = 294).

n = number of participants who responded

Lubumbashi is offered in the following specialized mental health services: the Neuropsychiatric Center '*Docteur Joseph Guillain*' of Lubumbashi (which has one neuropsychiatrist and one mental health nurse), the Neuropsychiatric Service of the Provincial Hospital Jason Sendwe (which has two neuropsychiatric interns) and the Neuropsychiatric Department of Lubumbashi University Clinics (which has one psychiatrist and one neurologist). These hospitals are private (not-for-profit) and public secondary and tertiary facilities, respectively, which can provide more complex treatments, including inpatient services during a mental health crisis. However, access to these facilities seems very limited due to the very high costs of the provided services. According to the activity reports consulted, in these health services, whose capacity for psychiatry services varies between 40 and 80 beds, 80% of the pathologies treated are mental disorders (depression disorders, mood disorders, manic episodes, delusional disorders, anxiety disorders, psychoactive substance abuse, puerperal psychosis, schizophrenia. . .) and 20% are neurological conditions (epilepsies, chronic headaches, pyramidal syndrome, somatoform disorders. . .). Nevertheless, the care packages offered in these hospitals have not been clearly described.

The minimum package of health services is offered in PHC centers of the health district of Tshamilemba. However, it is incomplete compared to the normative package, as it does not officially include mental health services. The content of this activity package is presented in Box 2.

## Box 2. Content of minimum package of activities currently offered in the PHC centers of the health district of Tshamilemba

The content of minimum package of activities currently offered in the health services of the health district of Tshamilemba consist of i) curative activities, including curative consultations, diagnostic services (ultrasound, medical laboratory), care for patients suffering from common diseases such as infectious diseases (malaria, typhoid fever. . .), trauma and violence, childbirth assistance and birth care, and management of chronic diseases (tuberculosis, HIV/AIDS, diabetes, arterial hypertension, etc.); general surgery activities; and referral and counterreferral activities, and ii) preventive activities relating to reproductive, maternal, new-born, child and adolescent health, vaccination, preschool consultation; health promotion activities such as family planning, behavior change communication, distribution of insecticide-treated mosquito nets; and rehabilitation activities such as the nutritional rehabilitation of children.

The interviewed health care providers said that they receive mental health cases for which they determine medical treatment. Epilepsy is the main condition and is treated with drugs such as diazepam and promethazine. Although it is classified as a neurological disorder, the providers consider epilepsy to be a mental disease. In addition, they have noticed that past attempts to treat patients with other mental health disorders (depression, psychoses. . .) by administering psychotropic drugs (amitriptyline, chlorpromazine) often failed, with patients returning to their family and community with unmet needs, showcasing that treatments for psychiatric illnesses are grossly lacking. Since then, they have decided to start referring them ex officio to provincial hospital Jason Sendwe, without offering any medical treatment. Due to the abovementioned limited financial accessibility, many patients resort to traditional healers and/or spiritual care interventions. The providers also expressed their fears regarding the mood instability and aggressiveness of some patients. From their responses, one could note the following:

*'We often wish to initiate the management of these cases, but we fear that it will be of poor quality. . . Currently, apart from epilepsy, which we treat well, we do not offer other mental health care for several technical, sociocultural and economic reasons. Patients who do not find their expectations met return home; some go to churches for prayer sessions, and others consult directly with traditional healers and diviners'* [FGD2].

According to the participants, patients attending PHC services in the health district of Tshamilemba are sometimes not taken care of for several reasons, including the following:

*'Lack of integration of the mental health care package [containing curative activities such as the identification of cases, curative consultations, diagnosis and medical care, neuropsychiatric emergencies, basic psychosocial care, psychological consultation, psychological first aid, and referral/counterreferral; preventive activities such as aftercare follow-up of cases, and social reintegration; and promotional activities including home visits, mental health awareness, psycho-education, reporting, monitoring and supervision] into the minimum package of activities, which gives providers no ability to perform adequate acts of care, no mental health information and training for providers, and no supervision or technical support from the National Mental Health Program; providers' poor perception of mental diseases; and the perception that mental health is a specific task of specialists'* [FGD3].

All participants (community participants and health care providers) expressed the need to see mental health care supply become available in primary care services because they believe that it is currently lacking. Participants in the study stated the following:

*'. . . we want psychiatric care to be integrated into health centers because when a family member is suffering from a mental illness, we often do not know where to take them. Sometimes we take them to a traditional healer or church for prayer. Either we give them medicine (paracetamol, diazepam, etc.) ourselves at home without success. . . or sometimes we [family] do not do anything. . . We are afraid to go to the provincial hospital (neuropsychiatry department) because the care there is very expensive'* [FGD5].

### Operational response capacity of the health district of Tshamilemba

The review of the national mental health system status report, consolidated district operational action plan and the activity reports of PHC centers showed that there is an absence of adequate and accessible community and hospital mental health services and specific community psychosocial support networks. It showed the absence of nonprofit organizations active in the field of mental health, as well as lack of associations of former patients or relatives of mental patients. The health areas have functioning health development committees.

Concerning mental health services, the qualitative data analysis showed that people with mental health conditions and/or psychosocial disabilities are not covered by formal health services. Community mental health care is not formally offered. Respondents mentioned that *'access to essential psychotropic drugs is very limited for people who need them'* [FGD2].

Analyzing the domain of mental health in the PHC, participants stated the following:

*'Mental health is not integrated into the minimum package of activities of all the PHC services of the health district of Tshamilemba, and there are no formal links between PHC providers, psychiatric specialists and informal psychosocial care providers such as traditional healers'* [FGD5].

It is not currently possible to receive treatment for mental health conditions in primary care.

Participants of the health district management team declared the following: '*The Haut-Katanga provincial health division does not have operational mental health coordination led by a focal point who can oversee the integration of mental health activities in operational action plans. The primary health care supervisor of the health district of Tshamilemba does not currently include the mental health component in his activity package*' [FGD3]. They said there is a need to '*strengthen the capacities of primary care providers on psychosocial and mental health care; supply health services with essential psychotropic drugs; harmonize diagnoses, case management and information tools; create links between primary care providers and mental health specialists; and provide health services with transport to escort patients to the psychiatric ward of provincial hospital Janson Sendwe*' [FGD3].

During the interviews, health care providers stated the following: '*Our capacities for the identification and management of mental disorders, mental health care and the provision of psychosocial support are limited to what we acquired during our school or academic training*' [FGD1].

The district health services are financed mainly by subsidies from the Congolese state, their own income, support from partners and rare donations. The Congolese state pays salaries and bonuses; however, these are insufficient to cover the district's daily needs. Analysis of the country's 2022 finance bill indicates that the overall national budget was US$9.9 billion (US $1 = CDF 2 093). Yet, the health sector budget forecast documents we analyzed revealed that the share of the national budget allocated to the health sector in 2022 was US$225 million. Of this amount, the share specifically allocated to mental health care was US$39 000. However, this budget is used mainly for the payment of salaries and the functioning of the central level of the National Mental Health Program (NMHP), and it is considered very insufficient to carry out activities at the operational level. Respondents indicated that '*patients [e.g., epileptics] followed at the Tshamilemba PHC center pay for their care out-of-pocket since they are not covered by any social protection system. This constitutes a barrier to access to care for other mentally ill patients*' [FGD1].

Regarding public awareness raising and linkages with other sectors, participants said, '*We do not do mental health promotion in our district because we do not know what to tell people, and there are no nonprofit organizations active in the field of mental health in our district*' [FGD2].

## Discussion

The results of this study, which aimed to analyze the existing demand and supply of mental health care in the urban health district of Tshamilemba and to examine the district's operational response capacity to address mental health show that the situation of mental health is worrying and that response capacities are weak.

This study highlights that the burden of mental health problems is significant in this urban area. In the household survey, half of the respondents reported mental health problems either themselves or in their family. Documentary analysis showed that in 2020, 5.3% of all general consultations in PHC centers in Tshamilemba concerned mental health. According to the 2019 Global Burden of Disease study [33], the prevalence of mental disorders including substance use disorders in the DRC is estimated at 13.23% of the population, i.e. 11.42 million people. This prevalence is likely to be underestimated because except for the provincial health divisions of North Kivu and South Kivu which are currently connected to the District Health Information Software 2 (DHIS2), in the other provincial health divisions, mental health data is

not routinely notified across primary care health services. Moreover, the epidemiological burden would probably be much heavier if population surveys were carried out, because the people in Congo generally do not consult for minor mental health conditions. There are currently no specific survey data available to determine the real extent of the situation. The present research precisely aims to filling this gap.

Surveyed participants stated that they and their family members were mostly affected by stress-related disorders, substance use and depression. There is however need for a more thorough investigation of the precise nature of the burden of mental health in the DRC in general, and in Tshamilemba district in particular.

The results indicated that only 1.4% of people with mental disorders had access to curative mental health care; thus sparking the debate on the very poor availability and accessibility of mental health care in the DRC. Based on this result, we can reasonably well assume that there is a significant mental health treatment gap in the Tshamilemba district, certainly when compared to the general figure of 10% of people with mental disorders accessing evidence-based treatments in LMICs overall [34].

Epilepsy is the most common neurological disease treated in PHC services in the health district of Tshamilemba. According to the standard classifications, epilepsy is a neurological disorder and not a mental health disorder [35]. However, in the Congolese context, it is considered a mental disease by health professionals and the community. This is a very interesting finding in itself. In addition, we believe that its perception by the population as a mental health disorder may mean that its successful clinical management could improve the confidence of care users in mental health services. This confidence would lead (potential) patients with mental health problems to accept and use the mental health care services offered. Furthermore, the findings pointed to the fact that epilepsy is currently not properly taken care of in the Tshamilemba district. Indeed, no effective anti-seizure medicines are available besides diazepam, which is used only for acute cases. The unmet need of epilepsy treatment is therefore considerable. The majority of people living with this neurological condition can unfortunately not access proper care. As in most LMICs where the treatment gap exceeds 75% [36], there is need to integrate epilepsy care in existing primary health care services.

Strengthening the technical capacity of health districts can reduce the burden of mental health problems [37, 38]. This includes improving mental health management by expanding the minimum package of activities of PHC services. This would allow patients to receive care in an open setting, maintain a connection with their families and remain productive, thus helping to reduce the related stigma [38]. In this regard, there is a need to either integrate traditional medicine practices and spiritual care interventions into primary care services or to collaborate with providers of such unconventional care in mental health provision.

However, once integration is understood as a technique that promotes the provision of mental health care only in hospital settings and by formal care providers, it is difficult to achieve, especially for traditional healers, because of problems related to medical beliefs, philosophies and practices [39]. Therefore, making integration an opportunity for training and coreferral patients, encouraging collaborative knowledge sharing, and extending invitations to traditional healers to treat specific diseases in hospitals are possible avenues [39].

According to the beliefs of the inhabitants of Lubumbashi, traditional African medicine remains the main recourse for treating mentally ill people. Research indicated that in the DRC, the first contact of mentally ill people with a health care provider is with traditional healers and/or spiritual workers (priests, lay pastors or imams), and that only as last resort modern health professionals such as psychiatrists and psychologists are contacted [40]. Also in the Tshamilemba district, traditional healers are the main source of care for mental health problems; future research into the effectiveness of their services remains however to be done.

The study showed that beyond the willingness of decision-makers to integrate mental health into PHC services, there is a strong lack of the necessary resources. Currently, in the city of Lubumbashi, there is only one neuropsychiatrist for the nearly 2 700 000 people. There are a few medical doctors in this city who are currently specializing in neuropsychiatry at the university clinics of Lubumbashi, but the exact number is not known. This problem of a lack of specialized human resources is a national one, since the DRC has a total of 102 neuropsychiatrists [23] per 89 561 403 populations, i.e., 1 psychiatrist per 878 053 people, although on a global scale, there are 1.2 psychiatrists per 10 000 people, with an overrepresentation of psychiatrists in high-income countries [39]. It should be noted that in LMICs, the scarcity of specialized human resources and their inequitable distribution between urban and rural areas mean that very few people with mental illness receive the mental health care they need [41].

Researchers [42] suggest (i) training or building capacity and delegating tasks to lay primary care providers, other lay health workers (social workers, traditional healers, spiritual care workers, etc.) and community actors such as community leaders and community health workers, family and community members. This delegation of tasks should be done in stages. It should follow the correct strategy of providers and nonproviders of care, (ii) ensure that primary mental health care also includes brief psychotherapeutic interventions, (iii) promote recovery-oriented community interventions for people with chronic disabling mental disorders, (iv) conceptualize training as a continuous process of strengthening clinical skills through supervision, (v) involve community partners in psychosocial interventions, (vi) integrate changes toward primary mental health care into broader health policy reforms, and (vii) promote intersectoral approaches to address the social determinants of mental health. Most of these recommendations are feasible in the Congolese context since they have been successfully tested in the health district of Lubero in the eastern region of the DRC [24].

Mental health legislation often reflects a commitment by countries at the highest level to protect people with mental illness and psychosocial disabilities from human rights violations [23, 37]. However, our results show that in the DRC, there is no mental health law. This means that alleged perpetrators of violations of the rights of psychiatric patients within families, communities and health services are not prosecuted. Thus, there is an urgent need for the country to accelerate the process of developing a mental health legislative framework that can open a pathway to the promotion of mental health and the rights of people with mental health problems and, by extension, equitable access to mental health care in a spirit of universal health coverage.

An important challenge remains to find a package for which all users of primary mental health care will be able to pay, given that 60–70% of Katanga's population lives on US$0.80 per day [43].

The lack of specialized services can be filled by the provision of primary mental health care as the first line of defense. However, the relationship between formal health services and traditional medicine and spiritual care structures seems to be difficult at present due to different beliefs about the origin of diseases, especially mental illnesses. This raises the question of whether it would be appropriate to envisage collaboration between these nonconventional health structures, operating on their own, and the primary care services, or to simply encourage collaboration with their facilitators by reflecting on a model of coordination with the primary care services [44].

Researchers believe that the strategy of integrating mental health into primary care is effective and can improve the quality of care at PHC services [45, 46]. However, such a strategy requires sufficient financial resources to establish and maintain a mental care offer in PHC services [38]. However, our findings showed that the state budget currently allocated to the mental health subsector, which is very insufficient, is mainly used only for the functioning of the

national office of the NMHP. It is important to reflect on the optimal mechanisms for financing mental health to avoid those initiatives taken by external partners suffering from sustainability [47].

To date, access to medication for psychiatric disorders and epilepsy is very limited in the health district of Tshamilemba. According to a 2022 country analysis [48], the lack of availability of essential antipsychotic and neuroleptic drugs, combined with their prohibitive cost, contributes to further worsen the treatment adherence.

Our findings revealed that access to counseling and psychotherapy was virtually non-existing in Tshamilemba district. These results corroborate those of the 2022 country analysis which affirmed that the possibilities of counselling and psychotherapy in the DRC are extremely poor [48]. However, some local or international non-profit organizations provide some level of support services for victims to sexual violence. Additionally, mental health and psychosocial support is a growing concern of humanitarian organizations, especially in conflict and post-conflict areas.

Stigmatizing attitudes vis-à-vis mental health problems are unfortunately (still) quite dominant in the community. These attitudes are related to the prevailing belief that mental disorders are linked to supernatural causes [20] such as witchcraft, sorcery, magic, bad spirits, bewitchment, transgression of taboos. . . These beliefs are largely shared by family members of people with mental health problems. Primary care providers acknowledge their limitations on how to address mental health problems. This contributes to leading people facing mental health problems to seek help from traditional healers and/or spiritual workers [49].

## Strengths and limitations

A first strength of our study is that, to the best of our knowledge, it is the first ever conducted in the city of Lubumbashi—and perhaps in the whole of the DRC—to analyze the demand and supply of mental health care, as well as the operational capacity of the health districts to address the huge unmet mental health needs of people. Our findings may inspire researchers, policy makers and health managers and constitute a basis for further reflection on the development of more evidence-based strategies to improve access to quality mental health care at the primary level.

A second strength of our study lies in its methodological set-up. The multimethod approach used, with triangulation of sources, methods and data, contributes to enhance the validity of our findings.

However, the levels of family and community participation in this study were low, which did not allow for an in-depth analysis of the role that these individuals play in the provision of mental health care. This is partly due to the context of the emergence of COVID-19, which is the period during which the current study took place. Additionally, missing from this study is the voice of traditional healers, although it is known that in the context of a lack of conventional medical care, these actors offer care. These aspects deserve to be considered in future studies.

Family and community members were a minority in the population included in the interviews conducted. Considering that these individuals may have little knowledge of the mental health issues being discussed, it is possible that they were influenced by the opinions expressed about health care providers. This was a limitation of the FGD variant (heterogeneous FGD) that we used. Indeed, soliciting perceptions from family and community members in the presence of health care providers could expose patients to a risk of response bias. This risk of bias was minimized by the style of moderation, which consisted of giving the floor to each participant and ensuring that everyone expressed themselves freely.

The classification of mental health disorders adopted herein almost follows the logic of the WHO's mhGAP intervention guide, which is not exempt from the risk of bias. Furthermore, the reluctance and hesitation of participants during this period when physical contact restrictions were in place could have increased the risk of over- or underreporting [50]. To minimize these latter biases, we selected interviewers who were familiar with the study context and who are also well known by community members.

## Conclusion

Our study established that the demand for mental health care in the Tshamilemba urban district in the city of Lubumbashi increased over time, but that, at the same time, the formal supply of mental health care remains poor. In Tshamilemba, traditional medicine remains the main source of mental health care provision. The current lack of evidence-based treatment results from a combination of poor operational capacity of primary care facilities, lack of resources, low priority of mental health care at policy level, and, last but not least, the reluctance of people with mental health problems to seek care given the high level of stigmatization of mental health issues.

The unmet mental health care needs are significant, and the responses to address them are grossly insufficient and inadequate. The need for urgent action in the area of mental health in this urban district, and undoubtedly throughout the city of Lubumbashi, is clear indeed. Among the 12 key shifts recommended by the WHO [51] to transform the mental health situation, three seem particularly relevant for our study setting: first, invest in universal health coverage policies that explicitly incorporate mental health care; second, raise community awareness for mental health problems and fight stigmatization and discrimination; and third, integrate a basic package of mental health care into existing primary care services. As Alegría et al. [52] point out rightly, such an ambition endeavor requires investing in the health workforce through task-shifting, training and supervision of primary care providers, delivering care close to where people live rather than expecting them to travel long distances to access residential psychiatric care. Our findings showed that young people suffered less than adults from substance use disorders and depression. Yet we generally know that young age is not a protective factor against these disorders. There is a need to explore why this would be the case in this district of DRC.

## Supporting information

**S1 File. Focus group discussion-interview guide.**
(DOCX)

**S2 File. Household survey questionnaire.**
(DOCX)

## Author Contributions

**Conceptualization:** Erick Mukala Mayoyo, Bart Criel, Yves Coppieters, Faustin Chenge.

**Data curation:** Erick Mukala Mayoyo, Didier Chuy.

**Formal analysis:** Erick Mukala Mayoyo, Didier Chuy.

**Funding acquisition:** Erick Mukala Mayoyo.

**Investigation:** Erick Mukala Mayoyo.

**Methodology:** Erick Mukala Mayoyo, Bart Criel, Joris Michielsen, Didier Chuy, Yves Coppieters, Faustin Chenge.

**Software:** Erick Mukala Mayoyo.

**Supervision:** Bart Criel, Joris Michielsen, Yves Coppieters, Faustin Chenge.

**Validation:** Erick Mukala Mayoyo, Bart Criel, Joris Michielsen, Didier Chuy, Yves Coppieters, Faustin Chenge.

**Visualization:** Erick Mukala Mayoyo, Faustin Chenge.

**Writing – original draft:** Erick Mukala Mayoyo, Bart Criel, Joris Michielsen, Didier Chuy, Yves Coppieters, Faustin Chenge.

**Writing – review & editing:** Erick Mukala Mayoyo, Bart Criel, Joris Michielsen, Didier Chuy, Yves Coppieters, Faustin Chenge.

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
