## [Decision Letter · Decision Letter 0]

16 Jan 2023

PONE-D-22-34666Mental health care in the city of Lubumbashi, Democratic Republic of the Congo: Analysis of demand, supply and operational response capacity of the health district of TshamilembaPLOS ONE

Dear Dr. Mukala Mayoyo,

Thank you for submitting your manuscript to PLOS ONE. After careful consideration, we feel that it has merit but does not fully meet PLOS ONE’s publication criteria as it currently stands. Therefore, we invite you to submit a revised version of the manuscript that addresses the points raised during the review process.

We look forward to receiving your revised manuscript.

Kind regards,

Taofiki Ajao Sunmonu

Academic Editor

PLOS ONE

2. Please provide copies of the interview guide and survey as supplementary files

Reviewers' comments:

Reviewer's Responses to Questions

**Comments to the Author**

1. Is the manuscript technically sound, and do the data support the conclusions?

Reviewer #1: Partly

2. Has the statistical analysis been performed appropriately and rigorously? 

Reviewer #1: Yes

3. Have the authors made all data underlying the findings in their manuscript fully available?

Reviewer #1: No

4. Is the manuscript presented in an intelligible fashion and written in standard English?

Reviewer #1: Yes

5. Review Comments to the Author

Reviewer #1: I wish to commend the authors for conducting and reporting on an interesting and original exploratory study on the highly relevant and neglected topic of mental health care in Africa. As pointed out by the authors, the burden of mental health in low- and lower middle-income countries is very high and remains largely unaddressed. The in-depth knowledge generated by field-based exploratory studies such as this one should prompt funding of further implementation research and service provision for mental health in resource-limited settings.

PlosONE criteria for publication:

1.The study presents the results of primary scientific research.

2.To the best of my knowledge, results reported have not been published elsewhere.

3.Experiments, statistics, and other analyses are performed to a high technical standard and are described in detail. The methodology and results of the household survey and the focus group discussions would benefit from additional detail.

4.Conclusions need some revision and are not entirely supported by the data in their current form. The results should be presented in more detail and put into context with additional references to existing literature. The paragraph limitations should also describe strengths.

5.The article is presented in an intelligible fashion and is written in standard English.

6.The research meets all applicable standards for the ethics of experimentation and research integrity.

7.The article does not fully adhere to the STROBE guidelines for reporting of cross-sectional studies, the COREQ criteria for reporting qualitative research, nor community standards for data availability.

General Review

•What are the main claims of the paper and how significant are they for the discipline?

This mixed-methods study explores the burden of mental health, and demand and provision of mental health care in one small health district of Lubumbashi, Democratic Republic of Congo, through a documents review, focus group discussions and a population household survey.

The study describes a high burden of mental health and unmet demand for mental health care, as well as extremely limited access and offer of mental health services in general, and especially at the primary care. The study highlights gaps between existing policy and non-existent practice with regards to integration of mental health care in primary care services. It highlights vastly insufficient financing, the absence of non-governmental implementation and funding partners, inadequate training of health workers, and insufficient knowledge about the causes and treatments of mental health problems in the community, leading to stigma and inadequate beliefs. Interestingly, the study also identifies epilepsy as an important illness perceived by health workers as a mental health problem and the lack of adequate treatment for this important neglected disease.

•Are the claims properly placed in the context of the previous literature? Have the authors treated the literature fairly?

The claims would benefit from additional references to place the results in the context of previous literature.

•Do the data and analyses fully support the claims? If not, what other evidence is required?

The data and analyses fully support the claims of unmet needs for mental health care. The detailed proposed solutions however are not fully supported by the data. The range of possible responses to the evidence produced by the study should be widened and the uncertainty as to their effectiveness should be acknowledged and supported by additional references to existing data.

•PLOS ONE encourages authors to publish detailed protocols and algorithms as supporting information online. Do any particular methods used in the manuscript warrant such treatment?

Yes. The questionnaires used for the focus group discussions and the survey should be made available, as well as a detailed algorithm outlining the cross-sectional survey.

•If the paper is considered unsuitable for publication in its present form, does the study itself show sufficient potential that the authors should be encouraged to resubmit a revised version?

Yes.

•Are original data deposited in appropriate repositories and accession/version numbers provided for genes, proteins, mutants, diseases, etc.?

The original data are deposited in an appropriate repository.

•Does the study conform to STROBE guidelines and the Fort Lauderdale agreement?

The study will need to be revised to conform to the STROBE guidelines for reporting of cross-sectional studies.

•Are details of the methodology sufficient to allow the experiments to be reproduced?

No. Additional detail is necessary on the questionnaires (FGD and survey) and the methodology of the survey.

•Is any software created by the authors freely available?

No.

•Is the manuscript well organized and written clearly enough to be accessible to non-specialists?

Yes.

•Is it your opinion that this manuscript contains an NIH-defined experiment of Dual Use concern?

No.

Line by line comments

Abstract:

Methods

46What type of survey was conducted?

48-51It is not clear how the burden of mental health problems was assessed, nor how demand was analysed, or how the district’s operational capacity was assessed.

Results

60Which methods led to the conclusion that traditional medicine is the main source of care?

Conclusion

64-67While I agree that integrating a basic package of mental health care into primary care seems to be relevant, it is not clear how the results support such a specific intervention. Mental health care seems to be lacking at all levels of the health care system and this study identified multiple barriers to access. Moreover, this study was conducted in one small health district, and one should be careful to generalize the findings to the entire country, or at least argue why the authors think the situation is similar in most areas.

Introduction

89Beds in outpatient facilities seem contradictory. Maybe inpatient facilities?

Methods

98-99Not clear. Earlier you mention that mental health care has been integrated in Guinea, and here you state that it is not widely integrated.

86-99 It seems you are conflating two issues into one: insufficient financing and lack of integration at primary care.

100Use of traditional medicine and other alternative approaches in a situation where no other mental health care is available seems a solution, albeit imperfect, rather than a problem.

118What does this DALY refer to? The burden of all mental health disorders in DRC or the burden due to suicide? How does this compare with other countries?

163Do these numbers refer to the health district of Tshamilemba or Lubumbashi? If Tshamilemba, does this mean there is one public PHC for 283 000 inhabitants? And 59 private clinics? If not, what are the numbers for Tshamilemba?

177In line 140 it is stated 10 000 population.

270It is not clear what this formula refers to and to measure which effect the sample size was calculated. Also, a reference is needed.

276-279 These numbers should be in the Results section

280 Survey questionnaires and focus group discussion guides should be made available in supplementary files and referenced.

Results

366Unclear. Is this the proportion of all general consultations that was for mental health care?

370Table 3. Population is stated to be 10,000 and also 12,000 above. Need to harmonize numbers. What does (=2/1; 4/2; 4/1) refer to; need to clarify in the legend. In general, this table would be clearer if you specify the numerator and the denominator.

374Specify numerator/denominator

405Table 4. Title needs to specify that these are results from the household survey. The age comparison is not particularly useful. I would suggest removing it from the table and only referring to overall mean difference in the text.

405Table 4. Which definitions did you use for the mental health problems categories? For example, what is the difference between stress & anxiety problems? How did you define neuroses and psychoses?

414Young age was (NOT has been shown to be) a protective factor. In general, young age is not protective for substance use disorder and depression. It may be an interesting question to explore what this would be the case in this district of DRC.

442The data you describe on epilepsy are very interesting and reflect the importance of epilepsy as a major neglected neurological condition, with high stigma and grossly inadequate management. The fact that it is considered a mental health condition and inadequately treated with diazepam and promethazine is telling.

447The available treatments for psychiatric illnesses also seem to be severely lacking.

51539000 USD seems an extremely low amount for a country of 90 million inhabitants. Can you reference the source and also add the total national budget and the total amount allocated to health care. This would add some necessary perspective.

Discussion

Some general comments on the discussion. It would benefit from highlighting the results more and placing them in the context of existing evidence. Topics on which presenting existing evidence would be useful include mental health burden of disease and needs, mental health care offer in DRC and Africa, traditional healers as mental health providers, burden and care of epilepsy, access to medication for psychiatric disorders and epilepsy, access to counselling and psychotherapy. Your findings highlight significant stigma, both from providers and from the general public, surrounding mental health. This should also be addressed in the discussion.

533This urban area. The study only provides data on one area.

540I would be very cautious to suggest specific causes such as coronaphobia given that the study did not uncover any detailed diagnostic data and that the diagnostic term of neurosis is outdated and rather vague in this context. Rather highlight the need for further research to characterize the burden of mental health in DRC.

544-551 I agree it is very interesting, although unsurprising, that epilepsy is considered a mental health disorder. I would highlight that at the moment epilepsy is not managed appropriately as according to the results there were no proper antiseizure medications available besides diazepam, which is used only in for acute seizures. Maybe it is also worth highlighting the enormous unmet need of epilepsy treatment worldwide, with more than 70% of people living with epilepsy not accessing care1.

599 Remove with.

621Title should be Strengths and Limitations. Please also highlight strengths of your study.

Conclusions

While I agree that a pilot project would a valuable response to the unmet needs in mental healthcare described in the study, this article is not the place to provide a detailed plan for such a project. Conclusions should be based on the results and propose future directions for research and public health action. I would suggest to shorten the conclusions, base them more closely on the results, and add additional references on the proposed responses.

6. PLOS authors have the option to publish the peer review history of their article (what does this mean?). If published, this will include your full peer review and any attached files.

Reviewer #1: **Yes: **Gilles Van Cutsem

---

## [Author Response · Author response to Decision Letter 0]

1 Mar 2023

No specific comments. We welcome comments and suggestions from the academic editor and reviewers.

---

## [Editor Report · Decision Letter 1]

16 Mar 2023

Mental health care in the city of Lubumbashi, Democratic Republic of the Congo: Analysis of demand, supply and operational response capacity of the health district of Tshamilemba

PONE-D-22-34666R1

Dear Dr. Mukala Mayoyo,

We’re pleased to inform you that your manuscript has been judged scientifically suitable for publication and will be formally accepted for publication once it meets all outstanding technical requirements.

Kind regards,

Taofiki Ajao Sunmonu

Academic Editor

PLOS ONE

Additional Editor Comments (optional):

Great works. Article merit acceptance with the submission of rebuttal to the comments made by the expert reviewers
---

## [Editor Report · Acceptance letter]

28 Mar 2023

PONE-D-22-34666R1 

Mental health care in the city of Lubumbashi, Democratic Republic of the Congo: Analysis of demand, supply and operational response capacity of the health district of Tshamilemba 

Dear Dr. Mukala Mayoyo:

I'm pleased to inform you that your manuscript has been deemed suitable for publication in PLOS ONE. Congratulations! Your manuscript is now with our production department. 

Kind regards, 

on behalf of

Dr. Taofiki Ajao Sunmonu 

Academic Editor

PLOS ONE